# Effects of Borax, Sucrose, and Citric Acid on the Setting Time and Mechanical Properties of Alkali-Activated Slag

**DOI:** 10.3390/ma16083010

**Published:** 2023-04-11

**Authors:** Peiqing Li, Deyong Chen, Zhirong Jia, Yilin Li, Shuaijun Li, Bin Yu

**Affiliations:** 1School of Civil and Architectural Engineering, Shandong University of Technology, 266 Xincun Road, Zibo 255000, China; 2School of Transportation and Vehicle Engineering, Shandong University of Technology, Zibo 255000, China; 3Shandong Jiuqiang Group Co., Ltd., Zibo 255000, China

**Keywords:** alkali-activated slag, sucrose, borax, citric acid, setting time, mechanical properties

## Abstract

The setting time of alkali-activated slag (AAS) binders is extremely short, while traditional retarders of Portland cement may be invalid for AAS. To find an effective retarder with a less negative impact on strength, borax (B), sucrose (S), and citric acid (CA) were selected as potential retarders. The setting time of AAS with different admixtures dosages of 0%, 2%, 4%, 6%, and 8%, and the unconfined compressive strength and beam flexural strength of 3 d, 7 d, and 28 d AAS mortar specimens were tested. The microstructure of AAS with different additives was observed by scanning using an electron microscope (SEM), and the hydration products were analyzed by energy dispersive spectroscopy (EDS), X-ray diffraction analysis (XRD), and thermogravimetric analysis (DT-TGA) to explain the retarding mechanism of AAS with different additives. The results showed that the incorporation of borax and citric acid could effectively prolong the setting time of AAS more than that of sucrose, and the retarding effect is more and more obvious with the increase in borax and citric acid dosages. However, sucrose and citric acid negatively influence AAS’s unconfined compressive strength and flexural stress. The negative effect becomes more evident with the increase in sucrose and citric acid dosages. Borax is the most suitable retarder for AAS among the three selected additives. SEM-EDS analysis showed that the incorporation of borax does three things: produces gels, covers the surface of the slag, and slows down the hydration reaction rate.

## 1. Introduction

Ordinary Portland cement (OPC) has been widely used in various types of construction due to its good mechanical properties. However, the production of OPC requires a large amount of energy and non-renewable minerals (e.g., limestone), and the process of calcination of limestone at high temperatures (1500 °C) emits large amounts of CO_2_ into the atmosphere [1,2,3,4]. In the context of global efforts to reduce carbon emissions and natural resource consumption, alkali-activated materials (AAMs) have attracted the interest of many researchers, and research on AAMs has been conducted for many years [5,6,7]. Alkali-excited materials are industrial wastes rich in calcium–silica–alumina, such as slag, fly ash, and metakaolin as precursors, which react slowly when reacting with water, but their activity is stimulated while obtaining better properties when in an alkaline environment [8,9,10]. In previous studies on the preparation of alkali-excited materials, mainly the precursors are mixed with alkaline solutions such as sodium hydroxide solution or liquid. The AAM obtained by this preparation method is called two-part AAM [11,12,13]. This preparation method has obtained more complete conclusions, and practical construction has been carried out in some countries. However, two-part AAMs require the treatment of highly concentrated alkaline solutions, which increases the safety risks during the production process. It is also inconvenient to store large quantities of alkaline solutions or alkaline silicate solutions because they tend to react with carbon dioxide in the air and thus reduce the alkalinity. The property that cement-based materials can be applied with only the addition of water poses a challenge to the further promotion of two-component AAMs [14]. In order to solve this problem, researchers found that the precursor and the excitation agent are mixed in advance, and only water is needed to obtain better performance. Compared with two-component AAM, one-part AAM has higher energy consumption and more complicated operation steps, but the production process is safer. As an environmentally friendly material that has received increasing attention recently, alkali-activated slag (AAS) has received much attention from scholars due to its low energy consumption and low CO_2_ emissions during the production process [12,15,16,17]. AAS has the advantages of rapid development of mechanical strength [18,19], excellent corrosion resistance [20,21,22], and fire resistance [23]. Compared to alkali-activated, low calcium fly ash needs to be cured under heating conditions [24], and AAS can show better mechanical performance when just cured at room temperature [25,26].

However, AAS shows a rapid setting characteristic when alkali metal silicate (nSiO_2_-K_2_O, K is any alkaline metal) is used as the activator [27,28,29], which has led to restrictions on the widespread use of AAS. Therefore, it is necessary to find a retarder that can effectively prolong the setting time of AAS. The retarding method of cementitious material can be studied from two aspects. Inhibiting the dissolution of raw materials and reducing the formation of hydration products [30]. However, the sodium silicate-activated slag performance shows that the hydration reaction is hard to inhibit because silicon-rich properties promote many C-S-H and C-A-S-H generations [28,31]. A lot of research has been done on retarders of cement-based materials, as researchers first attempted to use cement retarders directly to alkali-activated materials (AAM) [32,33]. 

When borax was used as a retarder for cementitious materials, it effectively prolonged the setting time of OPC and sulphate aluminate cement (CSA) [34,35]. Studies showed that incorporating borax reduced the pH of the CSA pore solution, which was considered to be the reason for the ability of borax to prolong the setting time of OPC as well as CSA [34]. Some other studies found that borates were able to replace the sulfate ion in AFt to form B–AFt, thus affecting the coagulation time and the mechanical strength of CSA [36]. When citric acid was used in OPC, citrate was found to adsorb on the surface of clinker to form a protective layer to retard the dissolution of clinker [37], while when citric acid was used in concrete, it was found to improve the compressive properties and durability of concrete at certain doses [38]. Sucrose has been considered a very effective cement retarder, and some researchers have suggested that the retarding mechanism of sucrose is mainly through influencing the nucleation of C-S-H, i.e., by inhibiting the formation of the necessary nucleation through the aggregation of calcium ions and from inhibiting the production of related hydration products such as C-S-H [39,40]. 

Many research results show that alkali metal halides [41], phosphate/phosphate [41,42], and carboxylic acids/carboxylates [41,43] are very desirable cement retarders. Those retarders can be used for AAM retarding, but there are certain limitations. For example, calcium chloride used for AAS shows a significant retarding effect only at a certain concentration. When the concentration is low, calcium chloride accelerates the setting time, while in the long run, too high a concentration of calcium chloride will corrode the concrete structure of AAS [41]. The same problem occurs when phosphate is used as an AAS retarder: the retarding effect of phosphoric acid on AAS only occurs after reaching a certain concentration, and the retarding effect is susceptible to the concentration of phosphoric acid or phosphate [31,44]. This makes it difficult to obtain the stable retarding effect of AAS by the above-mentioned retarders. When citric acid was incorporated into FA-AAM, it accelerated the setting time of FA-AAM [45]. But some reports showed that malic acid and tartaric acid, which have the same structure as citric acid, had a retarding effect on AAS [46,47].

Borax, sucrose, and citric acid have been used in previous studies on the retardation of AAM. However, they all focused on low-calcium systems based on fly ash or metakaolin, and studies on slag-based high-calcium systems still need to be completed. Borax could prolong the setting time when used in most low-calcium AAMs [48,49]. At the same time, it was also found that when borax was used in fly ash-based AAMs, a linear increase in AAM setting time could be found during the increase of borax dosing from 2% to 8% [50]. In addition to prolonging the setting time, borax was also reported to improve the strength of AAMs [50,51]. Some researchers using sucrose as a retarder for FA-AAM found that the incorporation of sucrose was effective in prolonging the setting time [46,52]. However, when sucrose was used for high-calcium fly ash, sucrose did not prolong the initial setting time but prolonged the final setting time at 1% and 2% of sucrose incorporation [53]. In addition, when citric acid was used as a retarder for FA-AAM, some researchers found that citric acid did not act as a retarder but as an accelerator of the setting at 1.5% and 2.5% admixture [45].

The results of these studies indicate that it is not feasible to use traditional cement retarders directly to AAS. The differences in the hardening mechanism between AAS and cement-based materials result in the use of retarders that can have a retarding effect in cement-based materials that may not work when used for AAS. The current research on AAS retarders has the following problems:The different hardening mechanisms of AAS and cement may make it difficult for retarders of cement-based material to perform as expected when used in AAS;Most of the retarders that have been proven to prolong the setting time of AAS are limited by factors such as solution concentration, and the retarding effect is unstable and may have adverse effects on the long-term performance of AAS;In the reports of the retarding effect of retarders on AAS, researchers have focused mainly on the changes in setting time and flowability and have not paid enough attention to the mechanical properties and the relationship between changes in mechanical properties and changes in microstructure.

To solve the above problems, this paper investigates the effects of borax, sucrose, and citric acid on AAS’s setting time and mechanical properties at different dosing levels. We use the SEM-EDS to observe the microstructure and hydration products of AAS under different conditions and explain their reaction mechanism. This paper hopes to find a retarder that can effectively prolong the setting time of AAS without causing significant damage to the mechanical properties of AAS among these three admixtures and get reasonable dosing levels.

## 2. Materials and Methods

### 2.1. Raw Materials

The raw materials used in the testing are shown in Figure 1a–e. The slag was produced by Fuheng Mineral Products Trade Co., Ltd., Lingshou City, Hebei Province, China. The chemical composition analyzed by XRF is shown in Table 1.

The solid activator is Na_2_SiO_3_·_9_H_2_O produced by Tianjin Dengfeng Chemical Plant (Figure 1a). The modulus ratio of Na_2_O to SiO_2_ is 1 ± 0.05. Analysis of pure white particles: the retarder is decahydrate borax (B)-(Na_2_B_4_O_7_·10H_2_O) (Figure 1c) produced by Tianjin Fuchen Chemical Factory. Sucrose (S)-(C_12_H_22_O_11_) (Figure 1d) produced by Tianjin Beichenfang Reagent Factory. Monohydrate citric acid (CA)-(C_6_H_8_O_7_·1H_2_O) from Tianjin Zhiyuan Chemical Plant (Figure 1e). The sand used for the test is ISO standard sand.

### 2.2. Testing Design

#### 2.2.1. Testing Grouping

The precursor used in this experiment is slag, the activator is sodium silicate with a modulus ratio of 1, and the dosage is 6%. Borax, sucrose, and citric acid were used to add 2%, 4%, 6%, and 8% of the precursor mass, respectively, and were compared with the samples without a retarder. The specific experimental groups are shown in Table 2.

#### 2.2.2. Water-Binder Ratio

The optimum water consumption for the plaster test was determined by the standard consistency test, which was carried out according to the *Highway Engineering Cement and Cement Concrete Test Procedure* (JTG 3420-2020). The mortar test’s water-binder ratio (w/b) is fixed at 0.5 according to the requirements of the related technical specifications [54].

#### 2.2.3. Sample Maintaining

Following the related technical specifications [54], the prepared AAS mortar samples were placed in a standard curing box for 24 h and then demolded. After demolding, in turn the specimens were put into the curing chamber for curing. The water surface height should be 2 cm higher than the top of the specimen. The temperature of the curing box should be 20 ± 1 °C, and the humidity should be >90%.

### 2.3. Testing Method

#### 2.3.1. Setting Time

According to the related technical specifications [54] and following the standard consistency test, the standard consistency of different admixtures was determined, and then the solid activator and admixture were dissolved in water in advance. The slag and solution were then placed in a stirring pot, and the time was recorded. At the same time, the slow stirring began at 120 s, stopped for 15 s, and was then quickly stirred for 120 s. The initial and final setting times were measured after mixing and molding every 15 min or less.

#### 2.3.2. AAS Mortar Strength

Following the related technical specifications [54], sodium silicate and admixtures were dissolved in water in advance, and then the mixed solution was prepared according to the ratio of w/b = 0.5. The slag was placed in a stirring pot and mixed with a solution. Standard sand was added during the mixing process. After the mixing was completed, the fresh mortar was layered and filled into a 40 mm × 40 mm × 160 mm triple test mold, and its surface was flattened after vibration. According to the maintenance standard in Section 2.2.3, the mortar strength test was realized using cement mortar compression and bending tester after 3 d, 7 d, and 28 d maintenance.

### 2.4. Micro-Analysis

SEM-EDS analyzed the microstructure and chemical composition. The instrument used is a QUANTAFEG250 field emission scanning electron microscope, produced by the FEI Company of the United States. Before SEM-EDS analysis, the selected sample should be immersed in absolute ethanol for 7 d to completely stop the hydration reaction and dry state. X-ray diffraction (XRD) was performed using a Diffractometer system XRD D8. The source was operated at a voltage of 40KV using Cu Kα radiation with a scanning range of 10–70°. The samples were scanned at a speed of 3°/min with a step size of 0.02° to obtain the data for quantitative analysis. TG-DSC uses the SDT650 integrated thermal analyzer made by TA, USA, with a temperature range of 30–900 °C, nitrogen atmosphere, and a heating rate of 10 °C/min.

## 3. Results and Discussion

The experimental results of different parameter samples are shown in Table 3.

### 3.1. Setting Time

In Figure 2, with the addition of borax, the initial setting time and final setting time of AAS are prolonged. The larger the dosage, the better the retarding effect. When the borax dosage is 2%, the initial setting time only increases by 41%, and the final setting time only increases by 33%, which has little effect. Nevertheless, when the borax dosage reached 4%, the final setting time was significantly prolonged. The final setting time of B4 increased by 166% compared to M0. When the borax dosage reaches 8%, the initial and final setting times are considerably prolonged. The initial setting time of B8 increases by 358%, the final setting time increases by 400% compared to M0, and its retarding effect is the most significant. This finding is similar to that of previous studies using borax for low-calcium AAM [47,48,49].

Figure 3 shows that sucrose did not significantly improve the setting time of AAS. Compared with M0, the initial setting time of S2 was not improved and the final setting time only increased by 13%. The retarding effect of S4 is relatively apparent: initial setting time increased by 25%, and final setting time increased by 33%. The initial setting time of S6 and the final setting time both decreased. The initial setting time of S8 did not increase, but the final setting time increased by 7%. This conclusion differs from previous studies that used sucrose for low-calcium FA-AAM, but when the sucrose admixture was 2%, we were able to find similar conclusions to previous studies [46,51], i.e., no effect on the initial setting, but the ability to prolong the final setting time.

In Figure 4, compared with M0, when the citric acid dosage is 2%, the initial setting time only increased by 8%, and the final setting time did not change. When the dosage reached 4%, the initial and final setting time were prolonged by 108% and 120%, respectively. When the dosage of citric acid was 6% and 8%, the initial and final setting time were significantly prolonged. Compared with CA4, the initial setting time of CA6 was prolonged by 472%, and the final setting time was extended by 515%, while the initial setting time of CA8 was lengthened by 640%, and the final setting time was prolonged by 794%. This differs from previous results on the use of citric acid as a retarder for FA-AAM [31], where the incorporation of citric acid did not act as an accelerating effect, but on the contrary, showed a significant retarding effect at higher dosing levels.

### 3.2. Mechanical Property

In Figure 5a, AAS’s 7-d unconfined compressive strength increased significantly with the increase in borax dosage. Compared with M0, the 7-d unconfined compressive strength of B2, B4, B6 and B8 increased by 14%, 18%, 23%, and 20%, respectively. Compared with M0, the unconfined compressive strength of B2, B4, B6, and B8 increased by 101%, 99%, 94%, and 101%, respectively from 3–7 d. In the 28-d unconfined compressive strength, B6 showed the maximum value which was 9% higher than M0 and can be found different from M0 in 7–28 d. Unconfined compressive strength increased by 17%, and the 7–28 d unconfined compressive strength of B2, B4, B6, and B8 is stable. 

Figure 5b shows that incorporating borax improved AAS’s flexural strength. Compared with M0, the 3-d flexural strength of B2, B4, B6, and B8 increased by 7%, 16%, 32%, and 42%, respectively. It can also be found that the 3-d flexural strength of B6 is 14% higher than that of B4; B8 showed the maximum flexural strength at 3–7 d, but its flexural strength was 1.3% lower than that of B6 at 28 d. It can be seen from the 28-d flexural strength that the flexural strength is better when the borax dosage is greater than 2%. Compared with B2, the 28-d flexural strength of B4, B6, and B8 increased by 5%, 8%, and 7%, respectively.

Figure 6a shows that the incorporation of sucrose harms the unconfined compressive strength of AAS. Compared with M0, the 3-d unconfined compressive strength of S2, S4, S6, and S8 decreased by 35%, 77%, 85%, and 88%, respectively. Through the 7-d unconfined compressive strength, it was found that S2, S4, S6, and S8 decreased by 24%, 69%, 85%, and 87%, respectively, compared with M0. Further, by observing the 28-d unconfined compressive strength, it can be found that compared with M0, the compressive strength of S2, S4, S6, and S8 decreased by 26%, 76%, 85%, and 88%. 

Figure 6b shows that with added sucrose, the flexural strength of AAS is lost. At the time of 3 d, compared with M0, the flexural strength of S2, S4, S6, and S8 decreased by 42%, 56%, 63%, and 74%, respectively. At the time of 7 d, compared with M0, the flexural strength of S2, S4, S6, and S8 decreased by 44%, 60%, 64%, and 77%, respectively. At the time of 28 d, compared with M0, the flexural strength of S2, S4, S6, and S8 decreased by 10%, 53%, 68%, and 78%, respectively.

In Figure 7a, the incorporation of citric acid harms the strength development of AAS, which increases with the increased dosage. At the time of 3 d, compared with M0, the unconfined compressive strength of CA2, CA4, CA6, and CA8 decreased by 30%, 58%, 86.7%, and 94.6%, respectively. At the time of 3 d, compared with M0, the unconfined compressive strength of CA2, CA4, CA6, and CA8 decreased by 30%, 58%, 86.7%, and 94.6%, respectively. At the time of 7 d, compared with M0, the unconfined compressive strength of CA2, CA4, CA6, and CA8 decreased by 13%, 33%, 70%, and 81%, respectively. At the time of 28 d, compared with M0, the unconfined compressive strength of CA2, CA4, CA6, and CA8 decreased by 20%, 41.3%, 68.9%, and 95.7%, respectively.

Figure 7b shows that the incorporation of citric acid harms the development of the flexural strength of AAS. At the time of 3 d, compared with M0, the flexural strength of CA2 CA4, CA6, and CA8 decreased by 16%, 60%, 70%, and 81%, respectively. At the time of 7 d, compared with M0, CA2’s flexural strength had been reduced by 25%, CA4’s flexural strength had decreased by 64%, and CA6’s flexural strength had been reduced by 71% and 80%. At the time of 28 d, compared with M0, the flexural strength of CA2 CA4, CA6, and CA8 decreased by 16%, 51%, 74%, and 83%, respectively.

### 3.3. SEM-EDS Analysis

#### 3.3.1. SEM-EDS Analysis of Borax, Sucrose, and Citric Acid

The microstructure of different borax dosages is shown in Figure 8. According to the SEM images, a small amount of ettringite appeared in B2 compared to M0, and the formation of the C-S-H gel was reduced. It can be observed that the surface of AAS is entirely wrapped by a layer of gel complex in B6; this is similar to the phenomenon observed by Oderji et al. after incorporating borax into FA-AAM [49], and the growth rate of ettringite is slower than that of B2. This indicates that these complexes greatly hinder the reaction of AAS with water. At the same time, due to the incorporation of borax, the Ca^2+^ in the solution decreases, further weakening the early hydration reaction of AAS. The formation process of these complexes is mainly as follows: firstly, borax is hydrolyzed to boric acid, and then boric acid reacts to form B(OH)^4−^ in an alkaline solution with a pH value greater than 11, and B(OH)^4−^ reacts with Ca^2+^ in slag to form Ca(B(OH)_4_)^2^. This type of complex is a product similar to a gel structure, which has a semi-permeable membrane, and a small amount of water can pass through it. Due to the fast hydrolysis rate of borax at normal temperature, the wrapping effect of this type of complex is pronounced, so it can effectively prolong the coagulation time of AAS. With the gradual progress of the AAS reaction, this layer of the complex is gradually destroyed by hydration products so that the hydration reaction of AAS tends to be normal. The specific reaction equations are shown in (1)–(3).
(1)$$(B_{4}O_{7}{)}^{2-}+7H_{2}O\to 2OH^{-}+4H_{3}BO_{3}$$
(2)$$H_{3}BO_{3}+OH^{-}\to B{\left(OH\right)}_{4}{}^{-}$$
(3)$$2B{\left(OH\right)}_{4}{}^{-}+Ca^{2+}\to Ca{\left(B{\left(OH\right)}_{4}\right)}_{2}$$

Thus, it appears that the main reason for the retardation of AAS by borax is that the complex hinders water contact with the surface of slag particles to some extent after gradually wrapping them. Due to the rapid hydrolysis of borax, this wrapping effect appears very quickly, and this is also the reason for the increase in AAS mortar strength after the incorporation of borax. It was found by EDS analysis (Figure 9) that the presence of calcium, silicon, and aluminum in the sample was consistent with the image of C-S-H/C-A-S-H gel. This shows that the incorporation of borax does not affect the formation of the main hydration products in the later stage of AAS hydration and also proves that the complex and the hydration products can coexist. At the same time, this complexity makes AAS’s microstructure more uniform and dense [49,50]. This phenomenon is macroscopically manifested as AAS’s unconfined compressive strength and flexural strength after adding borax.

#### 3.3.2. Influence Mechanism of Sucrose

Figure 10a–c illustrates the microstructure of AAS with different sucrose dosages and the gradual emergence of micro-cracks in the microstructure of AAS with increasing sucrose dosages. However, the cracks are not apparent when the dosage is 2%, but when the dosage reaches 6%, the cracks can be clearly observed and exist in large quantities. At the same time, it was found that the hydration products of AAS, especially C-S-H and C-A-S-H, decreased by EDS analysis. The analysis results of SEM-EDS explained why sucrose did not affect the setting time of AAS and its strength damage. It can also be found that with the incorporation of sucrose, the C-(A)-S-H gel in AAS aggregates mostly in the form of small particles, which is considered to be similar to the poisoning phenomenon produced by its incorporation into cement [38,46], while a similar aggregation of small particles was also observed in SEM images of sucrose used in high-calcium fly ash [46].

#### 3.3.3. Influence Mechanism of Citric Acid

The microstructure of AAS with different citric acid dosages is shown in Figure 11. It can be seen from Figure 8 that the gel formed by AAS hydration gradually decreases with the increase in citric acid dosage, which is due to the addition of citric acid to neutralize OH^-^ in part of the solution and weaken the hydration reaction of AAS. This neutralization reaction prolongs the setting time of AAS to some extent. At the same time, the unconfined compressive strength and flexural strength of AAS are lost due to the reduction of hydration products.

### 3.4. XRD Analysis

Figure 12 shows that the admixture incorporation changed the crystalline phase of AAS hydration products, mainly C-(A)-S-H, to some extent. More traces of the presence of ettringite are observed in B6 compared to M0. Traces of the presence of Ca(B(OH)_4_)·2H_2_O are also found in B6, which is consistent with the phenomenon observed in SEM, and peaks pointing to the presence of calcite are also found in B6. In S6, it can be seen that the peaks that can point to the presence of calcite become less, while in CA6, no peaks that can point to the presence of ettringite are observed, while fewer peaks that can point to the presence of calcite are also found.

### 3.5. Thermal Analysis

Figure 13 shows that with temperature heating from 30 °C to 300 °C, M0, B6, and S6 showed significant mass loss. All showed significant heat absorption peaks between 110–120 °C, indicating the existence of water loss decomposition of C-(A)-S-H and water loss decomposition of calcium alumina in M0, B6, and CA6. Between 650 °C and 720 °C, M0, B6, and S6 CA6 all show significant heat absorption peaks, indicating carbonate decomposition. It is also noteworthy that the mass loss of CA6 was not significant during the heating of the temperature from 30 °C to 300 °C, and no obvious exothermic peak was found, which indicates that CA6 lost relatively little water at 30–300 °C, indicating that no obvious water loss of hydration products occurred.

## 4. Conclusions

Based on the study of the effects of different dosages of sucrose, citric acid, and borax on the setting time and mechanical properties of AAS, the following conclusions were made:The borax can effectively prolong the setting time of AAS, and this retarding effect is increased with the increase in borax admixture. It is also found that borax can enhance the flexural strength of AAS and keep the compressive strength stable. SEM-EDS analysis found that the borax admixture caused a semi-permeable film to form on the surface of slag particles, which slowed down the hydration reaction of AAS. By XRD, DT-TGA revealed that more C-(A)-S-H crystalline phases were generated, and traces of ettringite and calcite in borax were also found, which explains the ability of borax to enhance the flexural strength of AAS.The incorporation of sucrose only prolonged the final setting time at less than 4%. At the same time, AAS showed that the initial and final setting times were prolonged when the incorporation amount reached 4%. The retarding effect was no longer obvious when it was further increased. Meanwhile, C-(A)-S-H in sucrose was aggregated into small particles by SEM-EDS, and many microcracks were also found. The crystallinity of C-(A)-S-H was weaker than that of B6 by XRD and DT-TGA analysis, and traces of calcite and ettringite were also found.Citric acid prolongs the setting time by inhibiting the hydration reaction of AAS. However, it causes severe damage to the mechanical properties of AAS, so citric acid cannot be used as a suitable retarder in AAS. SEM-EDS pictures found that citric acid incorporation reduced the generation of C-(A)-S-H, while many microcracks were found on the surface of slag particles. By XRD, DT-TGA, it was found that the incorporation of citric acid deteriorated the crystallinity of C-(A)-S-H generated by AAS hydration, and fewer peaks were observed that could point to the presence of C-(A)-S-H, and no peaks were found that could point to the presence of calcium alumina. Meanwhile, no obvious decomposition of hydration products by water loss was found during heating, and only a small amount of carbonate decomposition was found.

## Figures and Tables

**Figure 1 materials-16-03010-f001:**
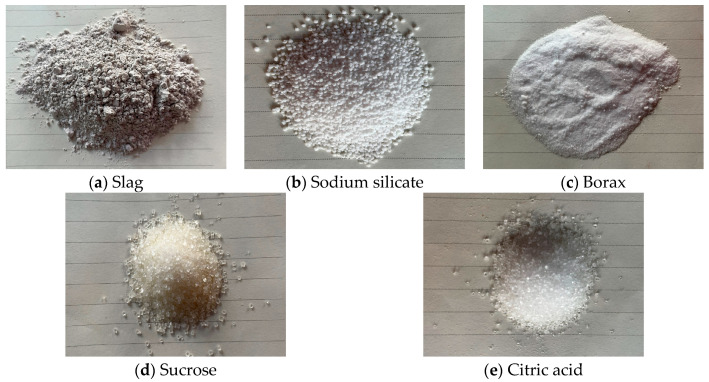
Raw Materials.

**Figure 2 materials-16-03010-f002:**
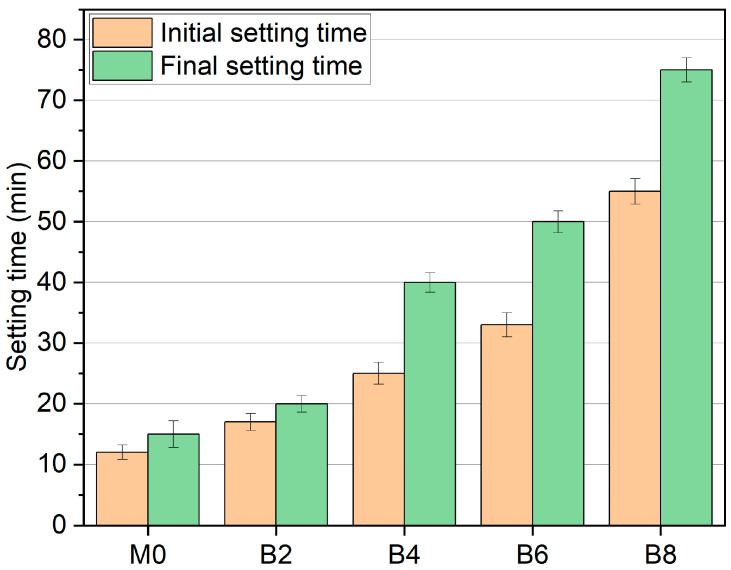
The effect of borax on the setting time of AAS with different borax dosages.

**Figure 3 materials-16-03010-f003:**
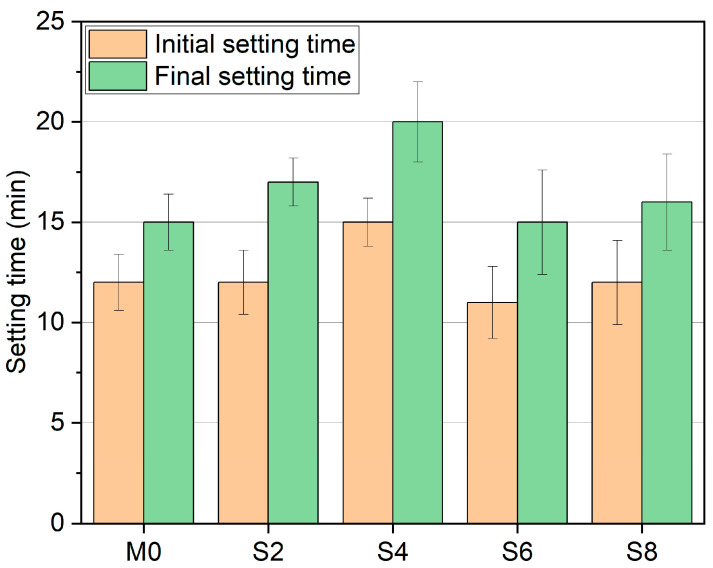
Effect of sucrose on AAS setting time under different dosages.

**Figure 4 materials-16-03010-f004:**
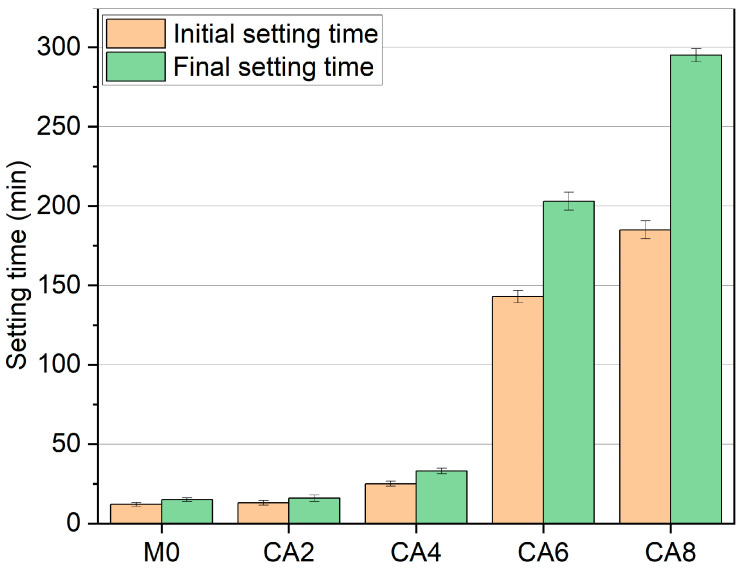
Effect of citric acid on setting time of AAS with different dosages.

**Figure 5 materials-16-03010-f005:**
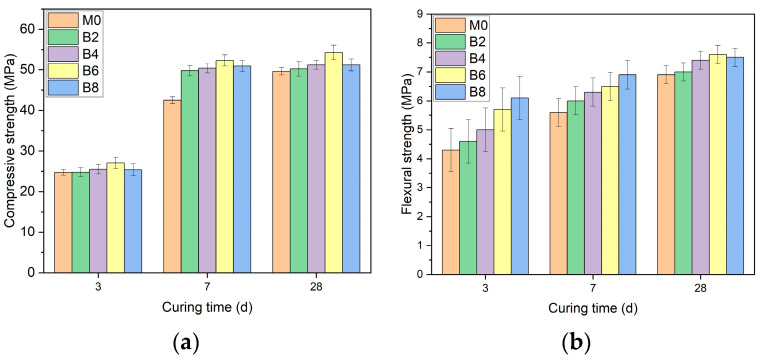
The effect of borax on the compressive and flexural strength of AAS with different borax dosages. (**a**) Unconfined compressive strength; (**b**) flexural strength.

**Figure 6 materials-16-03010-f006:**
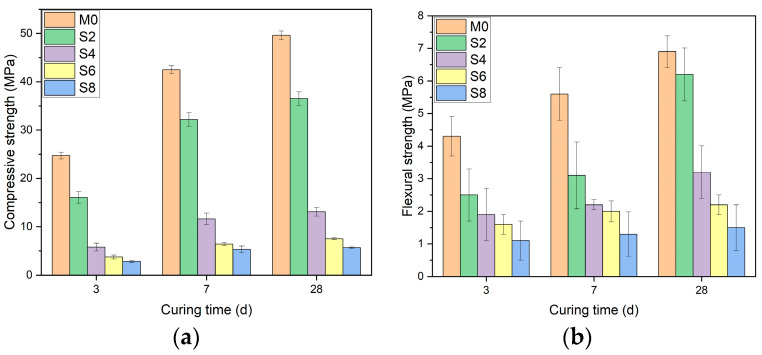
The effect of sucrose on the compressive and flexural strength of AAS under different dosages. (**a**) Unconfined compressive strength; (**b**) flexural strength.

**Figure 7 materials-16-03010-f007:**
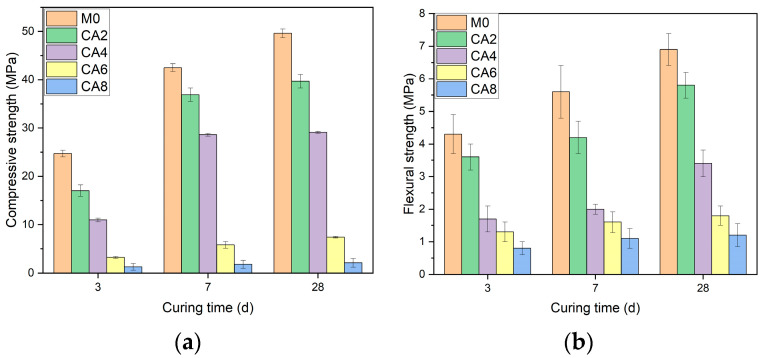
Effect of citric acid on compressive and flexural strength of AAS with different dosages. (**a**) Unconfined compressive strength; (**b**) flexural strength.

**Figure 8 materials-16-03010-f008:**
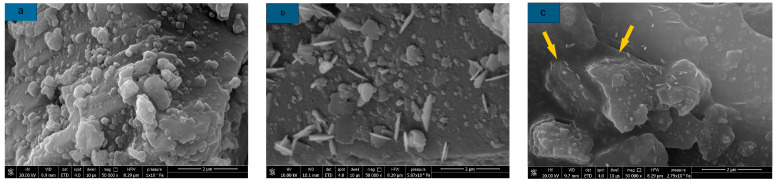
SEM images of samples with different borax dosages. (**a**) M0; (**b**) B2; (**c**) B6.

**Figure 9 materials-16-03010-f009:**
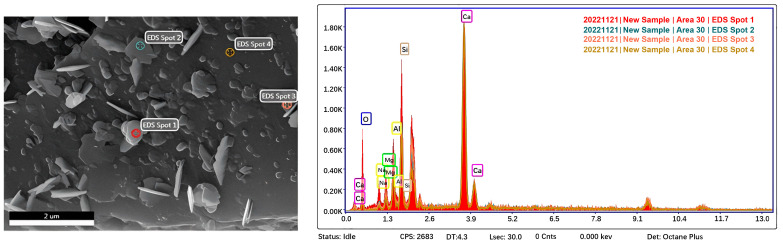
EDS images of B2 and B6.

**Figure 10 materials-16-03010-f010:**
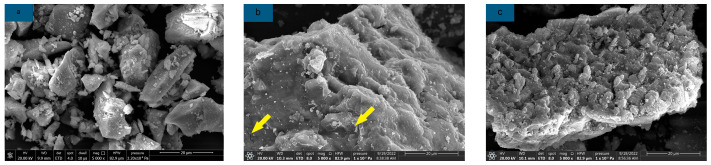
Microstructure of AAS with different sucrose dosages (**a**) M0; (**b**) S2; (**c**) S6.

**Figure 11 materials-16-03010-f011:**
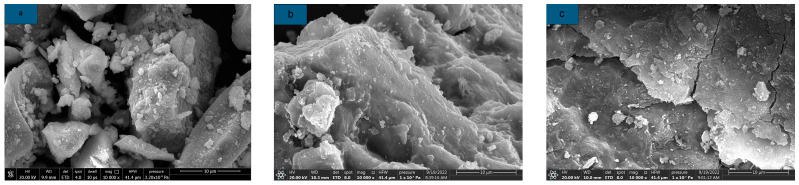
Microstructure of AAS with different citric acid dosages (**a**) M0; (**b**) CA2; (**c**) CA6.

**Figure 12 materials-16-03010-f012:**
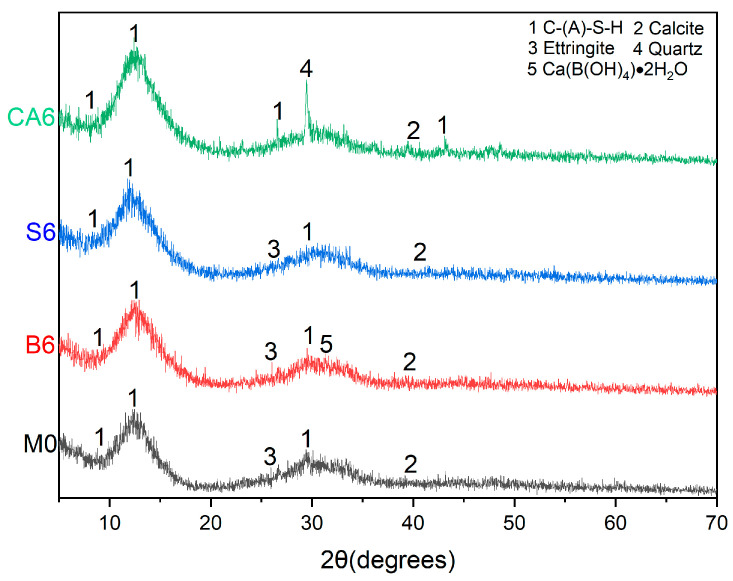
XRD patterns of M0, B6, S6, and CA6.

**Figure 13 materials-16-03010-f013:**
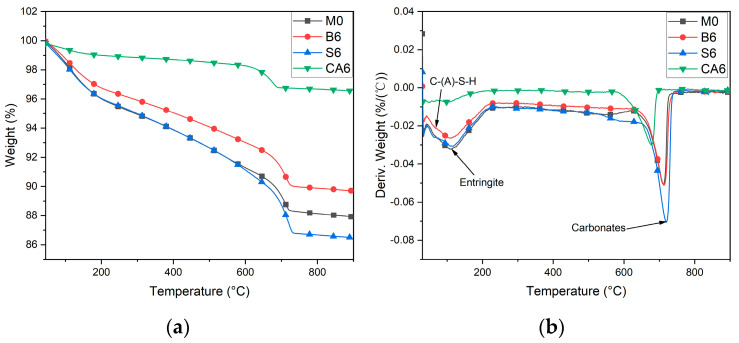
Thermal analysis of M0, B6, S6, and CA6. (**a**) Temperature-Weight Curves; (**b**) Temperature-Deriv Weight Curves.

**Table 1 materials-16-03010-t001:** Chemical composition of slag.

Oxide	CaO	Al_2_O_3_	SiO_2_	MgO	Fe_2_O_3_	TiO_2_	SO_3_	MnO	Na_2_O
mass%	42.6	14.2	27.8	8.09	0.378	1.2	2.46	0.401	0.55

**Table 2 materials-16-03010-t002:** Testing groups.

Mix Id	Activator Percentage	Na_2_O Percentages	Admixtures and Percentages
M0	6%	3%	0
B2	6%	3%	Borax—2%
B4	6%	3%	Borax—4%
B6	6%	3%	Borax—6%
B8	6%	3%	Borax—8%
S2	6%	3%	Sucrose—2%
S4	6%	3%	Sucrose—4%
S6	6%	3%	Sucrose—6%
S8	6%	3%	Sucrose—8%
CA2	6%	3%	Citric acid—2%
CA4	6%	3%	Citric acid—4%
CA6	6%	3%	Citric acid—6%
CA8	6%	3%	Citric acid—8%

**Table 3 materials-16-03010-t003:** Testing results of different parameter samples.

Mix Id	Initial Setting Time	Final Setting Time	Compressive Strength (MPa)			Flexural Strength (MPa)		
			3 d	7 d	28 d	3 d	7 d	28 d
M0	12 min	15 min	24.2	42.5	49.6	4.3	5.6	6.9
B2	17 min	20 min	24.8	49.7	50.2	4.6	6	7
B4	25 min	40 min	25.5	50.4	51.2	5	6.3	7.4
B6	33 min	50 min	27	52.3	54.3	5.7	6.5	7.6
B8	55 min	75 min	25.4	50.9	51.2	6.1	6.9	7.5
S2	12 min	17 min	16.1	32.2	36.5	2.5	3.1	6.2
S4	15 min	20 min	5.8	11.6	13.1	1.9	2.2	3.2
S6	11 min	15 min	3.7	6.4	7.5	1.6	2	2.2
S8	12 min	16 min	2.8	5.3	5.6	1.1	1.3	1.5
CA2	13 min	16 min	17.1	36.9	39.7	3.6	4.2	5.8
CA4	25 min	33 min	10.1	28.6	29.1	1.7	2	3.4
CA6	143 min	203 min	3.2	5.8	7.4	1.3	1.6	1.8
CA8	185 min	295 min	1.3	1.8	2.1	<1	1.1	1.2

## Data Availability

The data presented in this study are available on request from the corresponding author. The data are not publicly available due to the need for confidentiality.

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
