# Peer review of "Effects of Borax, Sucrose, and Citric Acid on the Setting Time and Mechanical Properties of Alkali-Activated Slag"

_materials, 2023, doi:10.3390/ma16083010_

Round 1

Reviewer 1 Report

Comments and Suggestions for Authors:

The paper entitled “Effects of Borax Sucrose and Citric acid on the Setting Time and Mechanical Properties of Alkali-activated Slag” deals with an interesting and actual topic. The paper is well written. Below you can find my recommendation:

1.     Authors claim they prepared AAS mortar samples, but no gradation and amount of used sand is reported. If sand was used, authors should include the information in the mix design description. If sand has not been used, authors should use the term paste instead of mortars.

2.     Most of the recent literature related on AAS is focused on the so called one-part mix, to avoid the use of liquid activator solutions that are caustic and dangerous to be handled [A-B]. Their behavior has been reported to differs from two-parts mixes. Is there a reason why they opted for liquid activators? The two types of mixes should at least mentioned.

[A] https://doi.org/10.3390/app10082865.

  [B] https://doi.org/10.1016/j.cemconres.2017.10.001

3.     All the values are without variance and graphs do not report error bars. It should be added and include in discussion where necessary.

4.     “confined compressive strength of B2, B4, B6, B8 only increased by 1%, 1.5%, 4%, 0.6%.” is not an increment especially if the error bars are not reported. I would state that compressive strength is steady.

5.     Correct all the chemical formulas by using superscripts and subscripts.

6.     The explanation of how the additives work is barely described only for borax and it is only supported by sem analysis. Further analyses (xrd, degree of reactions etc.. ) should be provided  for all the mixes to support the findings related to their mechanical properties. This part of the paper is weak and must be improved.

7.     Did the additives affect the workability of the mixes?

8.     Authors reported the presence of cracks in specimens produces using specific mixes. Did the authors measure the drying shrinkage of specimens? Because another big issue with AAS is the huge drying shrinkage.

Reviewer 2 Report

The topic of this research "Effects of Borax Sucrose and Citric acid on the Setting Time and Mechanical Properties of Alkali-activated Slag" is quite interesting and important for construction engineers. Authors discussed the influence of three retarders on setting times and mechanical properties with scientific evidence. Overall, this work is well organized, however, authors need to improve this work to enhance its readership. Please see my comments directly provided on the pdf file.

Author Response

请参阅附件。

Reviewer 3 Report

Fig 5, 6, 7 shall be redrawan, the readings can not be read
Fig.4 shall show error bars
Introduction is short shall be lengthened to state the novelty of the work
Consistency of AAS pastes shall be performed
were pastes or mortars produced... why author is mentioning mortar in the paper
there are no references in result and discussion section. Insert more references.
Why XRD and DT-TGA test not performed?

Reviewer 4 Report

The proposed work focuses on the effects of Borax Sucrose and Citric acid on the Setting Time and Mechanical Properties of Alkali-activated Slag. It is of potential interest to Materials journal readers.

Despite the importance of the subject addressed, this work needs many improvements to be ready for the publication in the Materials journal. 

Specific points of improvement :

-        The objective of this research must be more developed by a comparison with previous researches results.

-        All test standards must be indicated in Materials and Methods section

-        Explain why how the three retards can affect the setting time of ASS.

-        Justify the chosen additive used ratio.

-        Results need an in-depth discussion.

-        Quality of figures must be improved.

-     The conclusion section is too wordy. Only the main results must be indicated.

Round 2

Reviewer 1 Report

Authors improved the introduction and added thermal analyses and xrd. However peaks attribution is wrong. Usually C-A-S-H main peak overlaps calcite one at around 30° 2 theta and it is weakly crystalline. It is more reasonable that those peaks belong to quartz and arise from sand. Usually xrd and thermal analyses are carried out on pastes to avoid these problems. When samples are prepared from mortars.. only powder passing to 60 microns sieve should be used for analyses.. xrd should start from 5° to highlight the presence of ettringite... Please recheck

Reviewer 2 Report

Authors have improved their manuscript and addressed all my comments. Paper can be accepted in current form.

Author Response

Thank you for your valuable comments and suggestions!

Reviewer 3 Report

section 2.3.3- Omit 1450 g here in the text
its MPa not MPa
Put error bars in all bar charts showing setting time or consistency.

Reviewer 4 Report

The proposed work focuses on the effects of Borax Sucrose and Citric acid on the Setting Time and Mechanical Properties of Alkali-activated Slag. It is of potential interest to Materials journal readers.

I think that the revised version of the submitted paper is well improved by considering the reviewers and editor recommendations and remarks. Indeed, I think that this paper is accepted in this form.

Author Response

(The authors gave the same response as above.)
